# A chlorophyll halo over Maud Rise in the Southern Ocean

Bertrand Ducrocq [1] ✉, Nadine Steiger [1], Marcel du Plessis [2], Jean-Baptiste Sallée [1], Sébastien Moreau [3] & Sebastiaan Swart[2,4]

Phytoplankton blooms above the seamount Maud Rise in the Antarctic Ocean have been reported but their emerging mechanisms and their importance for the wider Southern Ocean are not well known. We use satellite data spanning over the last two decades and in-situ data collected from a ship, an underwater glider and Biogeochemical-Argo profiling floats to understand the processes involved in the formation of Maud Rise phytoplankton blooms. We find that the seamount generates upwelling of warm deep water that transports heat, and likely dissolved iron, to the surface via diapycnal mixing. This creates a recurring annular structure of chlorophyll concentration (or chlorophyll halo) in correspondence with the previously observed warm water and sea ice halo over Maud Rise. The in-situ observations reveal integrated chlorophyll-a concentrations of up to 100 mg·m$^{-2}$, which suggests exceptionally high phytoplankton biomass within the Southern Ocean, thus making the seamount a regional phytoplankton hotspot.

Southern Ocean primary production is known to be limited by a lack of iron and light[1,2]. On the contrary, the Southern Ocean is comparatively rich in macronutrients (nitrate, phosphate, and silicate), and so the processes that enhance iron supply to the surface layer have the potential to increase primary production. Annual phytoplankton growth in the subpolar Southern Ocean starts typically in spring when sea ice retreats, thus leading to an increase in sunlight exposure of the ocean surface[3]. At this time of the year, iron is typically available in the photic zone, having been resupplied by processes such as deep winter mixing[4], bacterial remineralization[5], and sea ice melt[6]. Phytoplankton quickly takes up iron hence their accumulation is typically observed in the vicinity of iron sources[7–10]. For example, enhanced primary production is often found around topographic features, where ocean currents such as the Antarctic Circumpolar Current, interact with the sea floor topography[3,11].

Maud Rise is a 3300 m tall seamount isolated in the 5000m-deep abyssal plain of the eastern Weddell Sea. It is located within the westward flowing southern limb of the Weddell Gyre, near 65S, 2E (Figure S1). The scientific interest of oceanographers for Maud Rise stems from the occasional occurrence of an open ocean polynya, first documented as the large-scale Weddell Polynya in 1974–1976[12] and more recently the smaller Maud Rise Polynya[13]. These rare polynyas are associated with deep ocean convection, which mixes warm and nutrient rich subsurface waters to the surface, thereby melting sea ice and likely enhancing primary production[12,14–18]. These openings within the sea ice have typically been explained by the upwelling or entrainment of deeper warm waters, such as those associated with the Maud Rise ocean current-topography interaction[19,20], Ekman effects[13,21–26], as well as atmospheric drivers through anomalous storm events[27] and positive air-sea heat flux[28,29].

According to the Taylor-Proudman theorem, the water column above a seamount is stiffened by Earth's rotation which effectively traps the water masses within the column and blocks any incoming flow as if the seamount extended up to the surface[24,30,31]. This concept known as Taylor column is a simple theoretical model, but describes well the Maud Rise ocean dynamics. The hydrography of the stagnant water masses overlying the seamount can significantly differ from the incoming flow from the surrounding Weddell Sea[15]. The Maud Rise Taylor column is associated with upwelling of warm and saline Warm Deep Water (WDW)[32,33] along the flanks of the Rise, which has been

[1]LOCEAN, Sorbonne Université/CNRS/IRD/MNHN, Paris, France. [2]Department of Marine Sciences, University of Gothenburg, Gothenburg, Sweden. [3]Norwegian Polar Institute, Tromsø, Norway. [4]Department of Oceanography, University of Cape Town, Rondebosch, South Africa. ✉e-mail: bertrand.ducrocq@ac-orleans-tours.fr

observed to leave an imprint on the sea ice cover, with a pattern of reduced sea ice concentration that follows the shape of the warm water halo[23,34]. The dynamics around Maud Rise and the associated polynya opening can also leave an imprint on the primary production, as described for the strong phytoplankton bloom that occurred over the Rise in 2017[17,18]. However, the link between the specific ocean dynamics associated with Maud Rise and the timing and intensity of the phytoplankton bloom developing over it remains poorly understood, while in-situ observations showing these linkages have been absent.

At the beginning of austral summer 2021-2022, a polynya occurred over Maud Rise before merging with the retreating sea ice edge (Fig. 1a). Following this polynya opening, we performed an extensive field campaign on board the icebreaker R/V *S.A. Agulhas II* between 6 to 16 January 2022 to observe the dynamics occurring in the Maud Rise area (Fig. 1b). During the measurement period, satellite images of chlorophyll-a (Chl-*a*) showed the occurrence of a phytoplankton

bloom over Maud Rise with a particularly high Chl-*a* concentration up to 1.6 mg·m$^{-3}$ on the flanks (Fig. 1b).

In this paper, we describe this phytoplankton bloom occurring over Maud Rise and discuss its spatial variability and frequency using 25 years of satellite observations of Chl-*a*, SST and sea ice cover. We also use a unique high-resolution glider survey of the Maud Rise, observations from Biogeochemical (BGC)-Argo floats and from the ship-based survey to gain a better understanding of the drivers of the halo-shaped phytolankton bloom. We then set the Maud Rise phytolankton bloom in the context of the wider Southern Ocean to highlight its importance.

## Results and Discussion

### A recurring halo-shaped phytoplankton bloom over Maud Rise

We use more than two decades of satellite data to assess the significance of the chlorophyll halo observed in January 2022 and to study

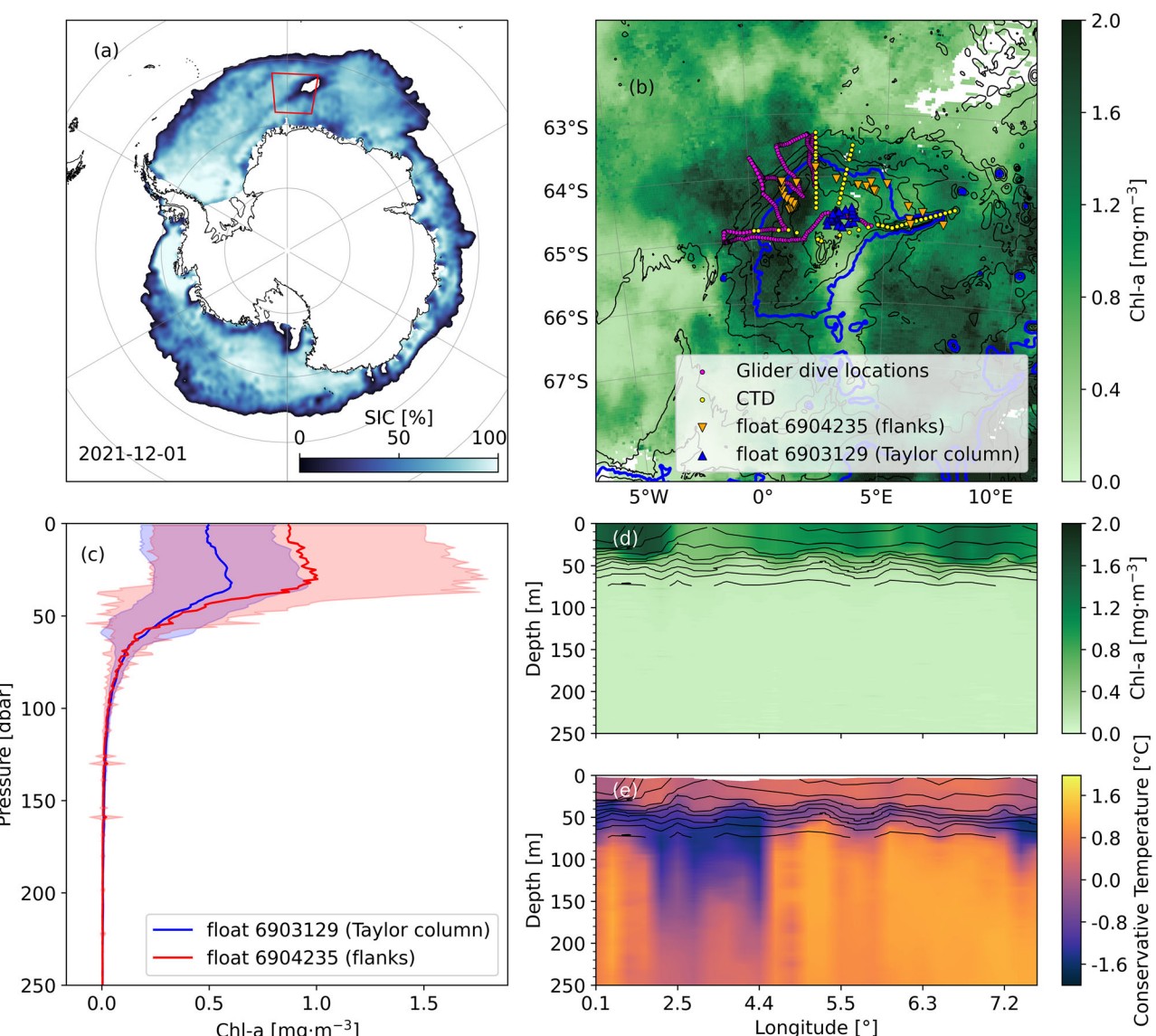

**Fig. 1 | Observations of the chlorophyll halo during the 2021-2022 austral summer. a** Sea Ice Concentration (SIC) on December 1, 2021. The red box delimits the study area over Maud Rise as in **b**. **b** Satellite Chlorophyll-a (Chl-*a*) concentration averaged over the cruise period (6-16 January 2022). Isobaths are in black and depicted every 500 m. The thick blue contour represents the 3500 m isobath. The marks are the locations of the glider (pink dots), CTD (yellow dots), and BGC-Argo floats (red and blue triangles) profiles. **c** Depth-profiles of the Chl-*a* concentration

from two BGC-Argo floats - float 6904235 in red and 6903129 in blue - that sampled across the Maud Rise flanks and inside the Taylor column, respectively. The average Chl-*a* concentration is plotted as a solid line while the filled area of the same color represents 1 standard deviation. Zonal section of the CTD measurements of Chl-*a* concentration **d** and conservative temperature **e** with isopycnals drawn as black lines every 0.1 kg·m$^{-3}$ between 1125.9 and 1126.6 kg·m$^{-3}$.

the phytoplankton bloom phenology in context with sea ice concentration and sea surface temperature. In the long-term mean, Chl-*a* concentration averaged over the Maud rise area (5W-10E, 62S-68S) starts to increase in late-November and reaches a maximum of 0.85 mg·m⁻³ in January (Fig. 2a). The year-by-year evolution of the Chl-*a* concentration indicates that the bloom over Maud Rise is a recurring feature and typical of the Southern Ocean Sea Ice Zone bloom phenology[3,35]. The onset of the phytoplankton bloom coincides with the retreat of the sea ice cover (Fig. 2a, c), followed by an increase in sea surface temperature (Fig. 2b). The Chl-*a* concentration reaches a maximum in the second week of January, when the sea ice has completely melted. The bloom reduces in intensity throughout late-January and February (Fig. 2a) and reaches a post-blooming phase in March, possibly due to increases in the grazing pressure or nutrient limitation[3,36].

The December-climatology of Chl-*a* concentration from 1998 to 2023 shows an annular phytoplankton bloom centered over the seamount (Fig. 2d). The Chl-*a* concentration reaches its highest magnitudes on the flanks, especially on the western and eastern flanks (≈0.85 mg·m3). The time evolution and the spatial distribution of Chl-*a* (Fig. 2a, d) thus suggest that a Chl-*a* halo over Maud Rise is a recurring feature that occurs almost annually during December (Figure S2). The average satellite observations of sea ice concentration and sea surface temperature show a low sea ice concentration halo shape as reported in Lindsay et al.[23] and a sea surface temperature halo[37] (Fig. 2e, f). The Taylor column overlying the seamount is observable by remote sensing of the sea surface height (Fig. 2g).

Focusing on the 2021–2022 austral summer, when we have high-resolution in-situ observations, the sea surface temperature and the fraction of area covered by sea ice are representative of the longer term conditions but the Chl-*a* concentration is higher than in other years (exceeding the standard deviation of the long-term mean in January) and peaks later by about a week (blue line in Fig. 2a–c). Overall, the observed Chl-*a* halo during 2021–2022 was a rather intense late bloom, with a maximum concentration of 1.48 mg·m⁻³ in January.

The seasonal cycle of the phytoplankton bloom shows strong inter-annual variability but moderate spatial variability, with recurring high biomass along the flanks of Maud Rise. The warm water pattern often matches the phytoplankton bloom pattern (Figures S2 and S3) which is supported by a positive correlation between the yearly time series of SST and Chl-*a* on the southeastern flank of Maud Rise (Pearson correlation coefficient *r* in the range of 0.4–0.8 with *p*-values under 0.05 - Fig. 3a). The correlation is significantly lower on the Northwestern flank (*r* between 0. and 0.2) probably due to eddy formation on this side of the seamount[30,37]. Therefore, we suggest that the inter-annual variability of the chlorophyll halo can be partly explained by the inter-annual variabil- ity of the warm water halo[38]. In addition, we suggest that anomalies in easterly winds may be responsible for enhanced upwelling in this part of the Southern Ocean and, therefore, lead to higher phytoplankton biomass as shown by Moreau et al.[10] in the region directly Southeast of Maud Rise.

A significant correlation between the SST and SSH yearly time-series was also observed, with values reaching up to r ≈ 0.6 on the Northeastern flank directly facing the incoming westward flow (Fig. 3b). These two correlations, between SST and SSH between and SST and the Chl-*a* concentration, hint at the pivotal role of ocean-topography interaction in the formation of chlorophyll halos as discussed in the next section.

## Linking phytoplankton biomass to ocean circulation dynamics

We used high-resolution underwater glider observations with ship-CTD and BGC-Argo float data to investigate the structure of the phytoplankton bloom together with the physical characteristics of the water column. The glider undertook seven crossings of the Maud Rise flank and through the Maud Rise halo. We analysed the first four glider

transects (white triangle, circle, square and diamond in Fig. 4a, b) when the phytoplankton bloom was ongoing (late November to early January). Both glider and CTD data (Figs. 4c, d, e, 1e and S4) indicate that the water column can, in a simplified view, be classified as a three-layer system structured from top to bottom. The first is an upper layer of relatively homogeneous temperature and salinity (≈ 50 meters) within which the mixed layer is contained. The upper layer is bounded below by the cold Winter Water (WW) layer[39], where the temperature remains below -0.7 °C. The third layer contains the warmer, saltier Warm Deep Water (WDW). The glider Chl-*a* and temperature sections (Figs. 4c and 4d) indicate that all the elevated Chl-*a* is contained within the upper layer. There is a decline of Chl-*a* concentration with depth starting between 40 and 50 m, consistent with the depth of the upper boundary.

The glider flank crossings show significant spatial variability, with higher Chl-*a* concentration (>1.5 mg·m⁻³) consistently found where the glider approaches the steepest topography of the Maud Rise flank, which is located around the 3500 m isobath. This isobath is used here as a reference depth throughout this study. The BGC-Argo floats further support the spatial distribution of Chl-*a* observed by the glider and seen in the satellite views, with float 6903129 in the Taylor column registering a substantially lower mean Chl-*a* concentration than float 6904235, which was bound to the Maud Rise flank (maximum recorded Chl-*a* values between 0 and 50 m for each float are 0.6 mg·m⁻³ and 1.0 mg·m⁻³, respectively; Fig. 1c). The ship-based CTD sections support a similar finding as shown in Fig. 1d and S4.

These sharp spatial gradients of Chl-*a* concentration are closely mirrored by changes in deeper ocean temperature and salinity (Fig. 4d, e), suggesting that the distribution of the surface layer Chl-*a* is closely tied to the deeper ocean fronts and circulation dynamics. In each dataset (ship CTDs, gliders and floats), the water mass distribution and its vertical extent changes abruptly over the Maud Rise flank, where the sea floor is approximately 3500 m deep. In particular, there is a sharp transition between the thick WW (up to 120 m thick) layer over the Maud Rise Taylor column, associated with cooler deep waters (below 0.4 °C), and the thin WW layer (20-40 m; Figure S5) bounded by warmer deep waters (≈1.2 °C) outside of the 3500 m isobath.

Such spatial patterns in WW thickness, as well as elevated warmer waters just below the WW at the flanks of Maud Rise suggest that ocean current-topography interactions lead to enhanced water mass interactions between the surface and deep water layers in these regions, leading to a response in phytoplankton production. Recent observations and modelling efforts have shown that the spatial pattern in upper ocean properties of the Maud Rise region are linked to the local bathymetry, with the flanks acting as a transition zone between the deeper waters of the Taylor column and the WDW of the surrounding ocean[20,38]. Steeply sloping isopycnals seen over the flank (Fig. 4d, e) may give rise to vertical transports, such as the upwelling of warm WDW through the WW and into the base of the mixed layer, as suggested through interleaving water masses observed by Mohrmann et al.[20]. Additionally, the sharper vertical temperature and salinity gradients at the base of the WW at the flanks would promote double-diffusive convection, such as salt fingering[40], thereby exchanging deeper water masses vertically to the upper ocean. These amplified oceanic vertical transports over the Maud Rise flank have the potential to supply iron from the iron-rich deep ocean to the notably iron-deficient upper sunlit layers[9,10], where it is made available to phytoplankton to photosynthesis, thereby leading to the higher values of Chl-*a* concentration that we observe as a more productive halo surrounding the Maud Rise.

We verified this iron supply hypothesis with the GLODAP database and its δ³He measurements throughout the Southern Ocean. 3-helium (³He) is released to the deep ocean by hydrothermal vents along with dissolved iron. There it can be transported thousands of kilometers before being upwelled closer to the surface to support primary

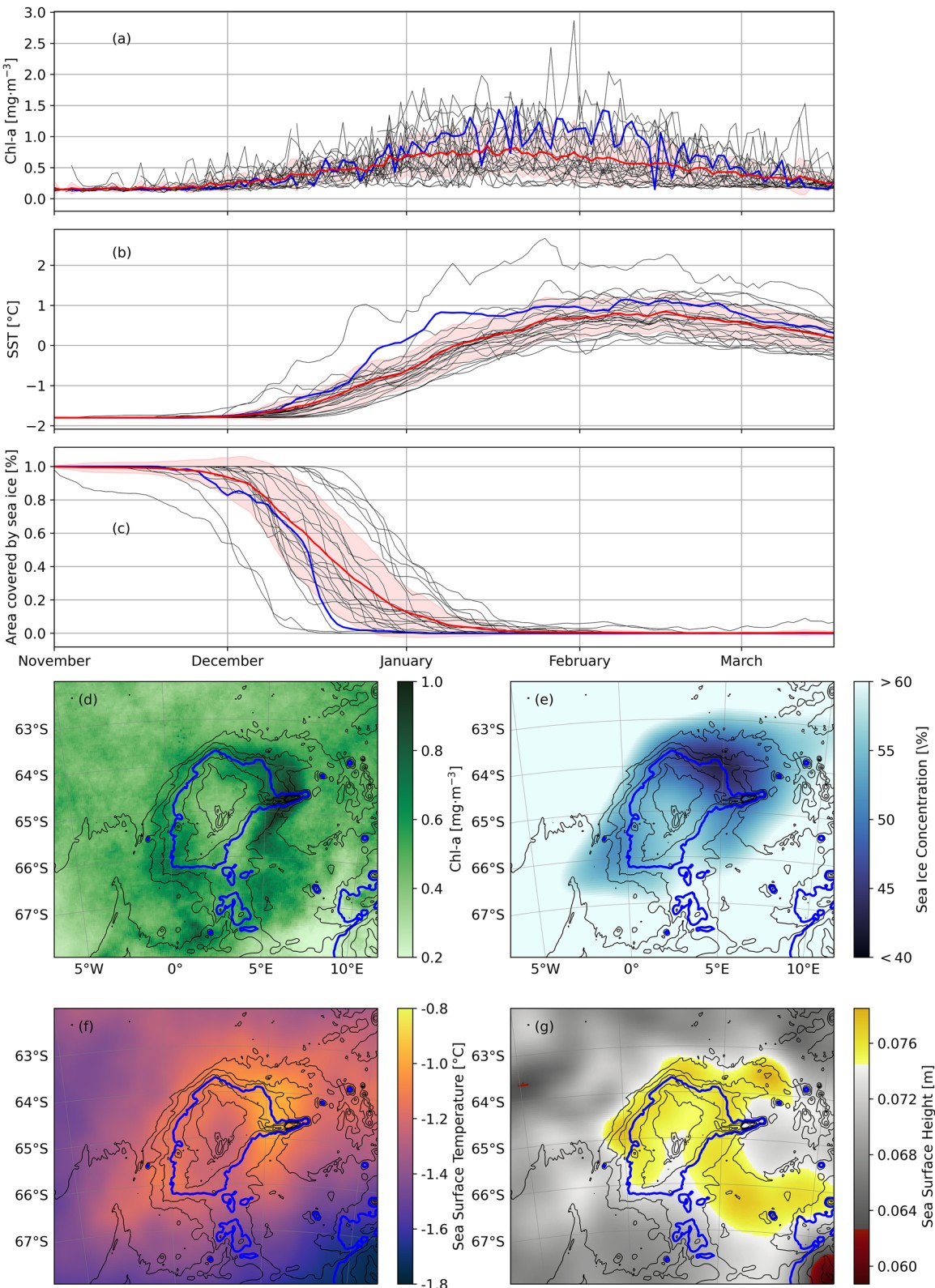

**Fig. 2 | Characterization of the halo structure from satellite observations.** Seasonal cycle of **a** Chl-*a* concentration, **b** Sea Surface Temperature (SST) and **c** fraction of sea ice cover over the area of the Maud Rise (5W-10E, 62S-68S). For each of those three variables, each year from 1998 to 2022 is plotted as a solid black line except the cruise year 2021–2022 represented as a solid blue line. The solid red line is the average over the period 1998-2022 and the red filled area represents the variation by one standard deviation. **d** December climatology (1998–2023) of the Chl-*a* concentration. **e** Climatology of sea ice concentration over the period mid-November to mid-December between 1998 and 2022. **f** December climatology (1998–2022) for SST. **g** January climatology (1998–2023) for sea surface height (SSH). The thick blue line in **d**–**g** represents the 3500 m isobath.

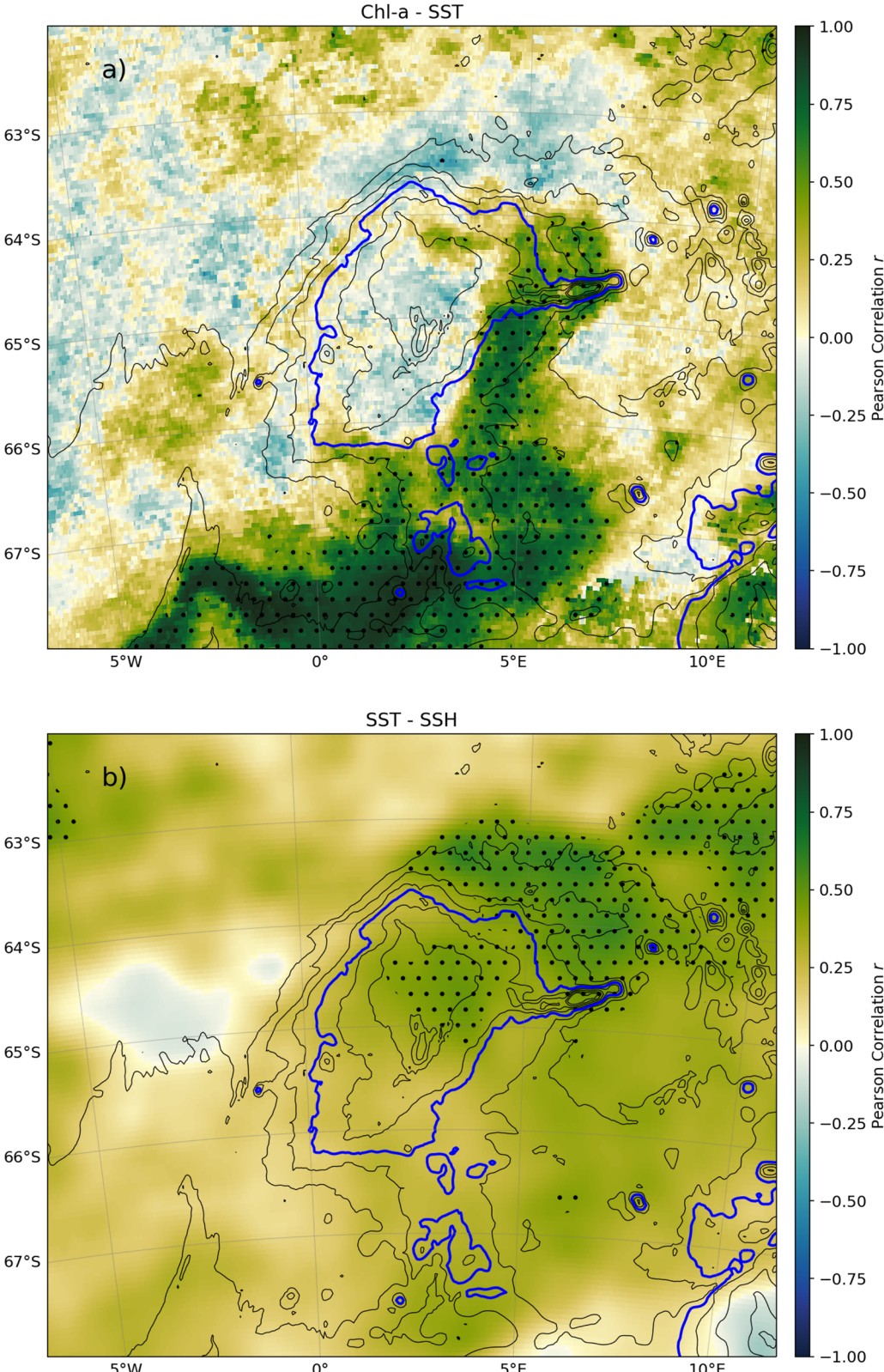

**Fig. 3 | Correlation maps of satellite variables.** Correlation maps between **a** SST and Chl-*a* concentration and **b** between SST and SSH. Isobaths, plotted as black solid contours, are drawn every 500 m. The 3500 m isobath is drawn in blue. Hatched areas show significant correlations ($p < 0.05$). See Figure S13 for more information on the slope of the regression.

production[9,41]. Therefore, measurements of $\delta^3$He can be used as a proxy for estimating the presence of hydrothermal iron. A transect of $\delta^3$He along the 0° meridian from 60°S to 70°S (Figure S6) gives a cross-section view of Maud Rise (approximately 65°S-66°S). We noticed two

$\delta^3$He anomalies above the flanks of the seamount (68°S-66°S and 65°S-64°S) in the range 200-900 meters depth (Figure S7). By looking at two additional transects of temperature and salinity, we find that the water masses, where the two $\delta^3$He anomalies are, have a salinity of

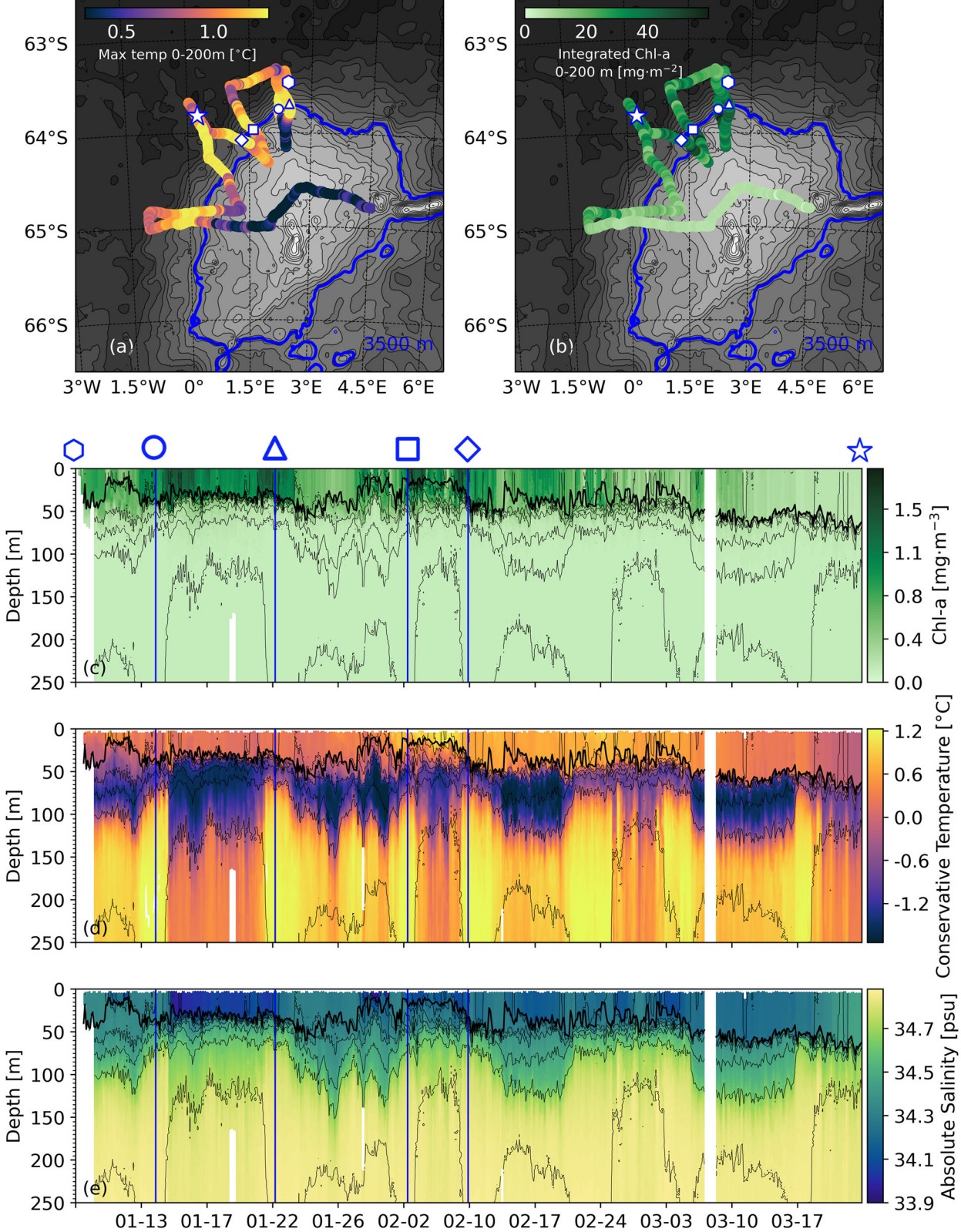

**Fig. 4 | High resolution views of the halo by an underwater glider.** (a) Maximum temperature observed between 0-200 m during the glider survey. (b) Glider-observed integrated Chl-*a* concentration between 0-200 m. For both (a) and (b), isobaths are depicted every 200 m and the 3500 m isobath is shown in blue. Blue lines in the glider sections of (c) Chl-*a*, (d) temperature, and (e) salinity correspond to checkpoints marked by the symbols in (a) and (b). Isopycnals are depicted every 0.1 kg·m$^{-3}$ between 1023.0 and 1030.0 kg·m$^{-3}$. The thick black line in (c), (d),(e) is the mixed layer depth (MLD) based on the 0.03 kg m$^{-3}$ density threshold criteria of De Boyer Montégut et al., 2004[50].

approximately 34.69 PSS-78 and temperature above 1 °C (Figure S7). These water masses are warmer, saltier, higher in $\delta^3$He and supposedly higher in dissolved iron concentration of hydrothermal origin than other water masses in the area (20°W-20°E,50°S-70°S) hence supporting the hypothesis of iron supply from the deep ocean (measurements of these water masses are within the black rectangle in Figure S8). It is also probable that some part of the iron suspended in the upper sunlit layers originates from sea ice melting[6].

### Relevance of the Maud Rise bloom for the Southern Ocean

Given the strong potential impact of the ocean dynamics over Maud Rise on the Chl-$a$ concentration, the question arises how strong the Maud Rise bloom is compared to the rest of the Southern Ocean. Based on the in situ observations, the depth-integrated Chl-$a$ concentration over the upper 200 meters reaches values up to 98.2 and 89.4 mg·m$^{-2}$ for Float 6904235 and CTD casts over the flanks of Maud Rise, respectively. Using all BGC-Argo floats[42] available for the period 2012-2020 in the Southern Ocean, we only find 39 profiles out of 8560 profiles in total (covering all areas of the Southern Ocean and all seasons), where the integrated Chl-$a$ over the upper 200 meters is larger than 100 mg·m$^{-2}$ (Figure S9). These 39 profiles do not include the 2 floats which were deployed over Maud Rise during the summer 2021-2022 to avoid any sampling bias.

Interestingly, the distribution of the locations with these exceptionally high Chl-$a$ biomasses in the BGC-Argo floats database are (1) in the notoriously productive Scotia Arc, (2) the sea ice zone of the Ross and Weddell Seas (including Maud Rise), and (3) a few locations near sub-Antarctic Islands in the Indian sector of the Southern Ocean where sedimentary iron is typically supplied by the inter- action of the Antarctic Circumpolar Current with shallow bathymetry[7].

The flanks of Maud Rise themselves account for 11 of these 39 profiles with exceptionally high Chl-$a$ biomass (ranging between 106.7 and 160.5 mg·m$^{-2}$ on the flanks of Maud Rise, compared to an average 132.6 mg·m$^{-2}$ for these 39 profiles). This adds evidence for the role of dynamic processes on the Maud Rise flanks leading to exceptionally high phytoplankton biomass accumulation in the Southern Ocean. This also suggests that Maud Rise may well be a critically important area for the Southern Ocean ecosystem and biological carbon pump[43,44].

### Concluding remarks and perspectives

We have shown that Maud Rise triggers annual phytoplankton blooms on the flanks of the seamount. The Taylor column above Maud Rise creates upward transport of warm water from the deep WDW layer towards the mixed layer. We propose that the WDW layer is enriched with iron that reaches the surface layer via frontal instabilities developing around the seamount, which fuels the phytoplankton bloom during the austral spring and summer. Other fingerprints of this oceanic upwelling includes a halo-shaped minimum in sea ice concentration in spring (mid November to mid December), consistent with subsurface heat impeding sea ice formation in this area, and enhancing melt rate. We also observe a halo-shaped maximum temperature around the flank of the Rise in December, which might be a consequence of an earlier sea ice retreat or a direct consequence of upwelled warm water. Our study suggest that the halo-shaped structures of sea ice concentration, temperature and chlorophyll are not independent but rather connected by the ocean dynamics stemming from the Taylor column of the seamount. As such, phytoplankton blooms over Maud Rise constitute an example of how marine ecosystems are shaped by mesoscale ocean dynamics and ocean circulation-topography interaction. More investigations are needed concerning the iron supply, the role of grazers on bloom dynamics, as well as the impact of atmospheric forcing on the strength of these blooms. Although trace metal nutrient measurements are complex to make, further trace metal measurements in open-ocean polynya regions, especially in the presence of shallow topography, would help elucidate the nutrient supply mechanisms between the deep ocean and surface layers. Understanding such dynamics and those elucidated in this study would help disentangle the bio-physical processes occurring in these complex polynya systems, with broader implications for Southern Ocean-wide primary production and carbon cycling.

## Method

### Satellite observations

The satellite observations used in this study are based on the "Global Ocean Colour Plankton and Reflectances MY L3 daily observations" dataset with daily and 4 km × 4 km resolutions, the "Global Ocean Gridded L 4 Sea Surface Heights And Derived Variables Reprocessed 1993 Ongoing" dataset with daily and 0.125° × 0.125° resolutions, and the "Global Ocean OSTIA Sea Surface Temperature and Sea Ice Reprocessed" dataset with daily and 0.05° × 0.05° resolutions. The three datasets extend from 1998 to 2022 (SIC, SST) or 2023 (Chl-$a$, SSH). The satellite-derived Chl-$a$ concentration is evaluated by observations in the visible domain of the ocean surface implying that cloud cover and sea ice hinder the retrieval of upper ocean Chl-$a$ concentrations[45]. The sea surface temperature and sea ice concentration estimates used in this study rely on the sea surface reflectance in the near infrared domain which is little affected by clouds but is subject to sea ice reflectance interferences and possible sensor calibration issues.

Based on the satellite-derived sea ice concentration, we define the fraction of the Maud Rise area covered by sea ice as the number of pixels with sea ice concentration above 15% in the domain (5°W- 10°E, 62°S-68°S) divided by the total number of pixels in the same domain. Chl-$a$ concentration derived from satellite imagery matches the measurements in situ (Fig. 1 b and 1d).

The correlation map (Fig. 3) was computed first by calculating the yearly times series of Chl-$a$ concentration and sea surface temperature (Figures S2 and S3) then computing the Pearson correlation coefficient map for these two time series. The slope of the regression of the Chl-$a$ - SST correlation was computed by multiplying the Pearson correlation with the ratio of the Chl-$a$ and SST standard deviations. Similarly, The slope of the regression of the SSH - SST correlation was computed by multiplying the Pearson correlation with the ratio of the SSH and SST standard deviations.

### Seaglider observations

A buoyancy-driven Seaglider (SG640) was deployed during the SO-CHIC cruise on January 10, 2022 on the flank of Maud Rise (63° 27.63 S, 02° 26.59E). Such gliders sample the physical and biogeochemical state of the upper ocean at high temporal and spatial resolution, with each V- shaped dive occurring between the surface and 1000 meters depth. The spatial resolution between profiles was 3.1 ± 1.5 km, while the mean temporal resolution was 3 hours. SG640 was equipped with sensors to measure salinity and temperature (SeaBird CT Sail), chlorophyll fluorescence at 695 nm and optical backscattering at 470 and 700 nm (WETLabs ECO Puck), and dissolved oxygen (Aandera optode 4831). SG640's last dive occurred on April 7, 2022. SG640 fluorescence was converted to Chl-$a$ and calibrated in the following way: (i) fluorescence units were converted to Chl-$a$ units (mg·m$^{-3}$) using the manufacturer calibration scale factor, (ii) the ship-board CTD was calibrated against in-situ bottle samples (Figure S10), and (iii) SG640 manufacturer-corrected Chl-$a$ was corrected to the calibrated ship-board CTD by finding the linear regression at the cast taken directly after the Seaglider deployment (Figures S11 and S12). The glider calibrated Chl-$a$ was subsequently corrected for non-photochemical quenching according to the methods in Xing et al.[46].

## CTD observations

We use ship-based CTD and Chl-*a* measurements that were taken at 58 stations over the Maud Rise. The stations were organised in 4 sections, two over the northern flank (Section N and Section NE) of which Section N aligns with the first glider section, and two sections that cross the Maud Rise from center to east and center to west, respectively (here shown as combined Section EW). The distance between the CTD casts is about 9 km and the depth alternatively 1000 m and full depth; for the center-west section, the distance is about 18 km and they cover the full depth. The rosette was equipped with 24 Niskin bottles for water samples, and the instrumentation was a SBE 9plus CTD Unit and SBE 11plus V5.2 Deck Unit. The initial data processing was done with the SBE Data processing software following the standard filters suggested for the SBE 9plus CTD unit. The conversion of fluorescence to Chl-*a* (mg·m$^{-3}$) was done using the bottle samples and a quenching correction as described in the previous paragraph. Isopycnals were calculated based on the Thermodynamic Equation of Seawater - 2010 via the gsw-python library that calculates in-situ density of seawater from absolute salinity and in-situ temperature.

## BGC-Argo floats observations

We use data from Biogeochemical-Argo (BGC-Argo) floats WMO6903129, deployed over the flank of Maud Rise at 64°32.860′ S, 03°12.533′ E and WMO6904235, deployed on top of Maud Rise at 64°31.0′ S, 1°30.0′ E. The float data are available at the Coriolis data platform. BCG-Argo floats provide insight into the water column physical state and submerge down to 1000 meters. They have no means of propulsion i.e. they drift according to currents, providing a quasi-Lagrangian view of the ocean. They are equipped with sensors for pressure, temperature, dissolved oxygen, conduc- tivity, Photosynthetically Active Radiation, Chl-*a* fluorescence, irradiance at 380, 412, 490 nm and backscattering at 700 nm. WMO6904235 data spans 29 descent-ascent cycles from 2021-12-11 to 2022-06-03 while WMO6903129 data spans 27 cycles from 2022-01-08 to 2022-05-22. We use the profiles of the ascent and corrected slope, non-photochemical quenching and the deep offset of Chl-a based on the backscattering data at 490 nm according to Xing et al.[47]. The processing is an adaptation of what was done in Schmechtig et al.[48]. The data are linearly interpolated onto a vertical grid of 1 db.

To produce Figure S9[49], we used floats observations collected and made freely available by the Southern Ocean Carbon and Climate Observations and Modeling (SOCCOM) Project funded by the National Science Foundation, Division of Polar Programs (NSF PLR -1425989 and OPP-1936222 and 2332379), supplemented by NASA, and by the International Argo Program and the NOAA programs that contribute to it (https://argo.ucsd.edu, https://www.ocean-ops.org). The Argo Program is part of the Global Ocean Observing System.

## GLODAP database

The figures S6, S7 and S8 were created with the ODV-online browser tool https://explore.webodv.awi.de/. Transects were contoured using ODV's DIVA gridding algorithm with a signal-to-noise ratio of 10.

## Data availability

The "Global Ocean Colour Plankton and Reflectances MY L3 daily observations" dataset is available at https://doi.org/10.48670/moi-00282. The "Global Ocean OSTIA Sea Surface Temperature and Sea Ice Reprocessed" dataset can be found at https://doi.org/10.48670/moi-00168. The "Global Ocean Gridded L4 Sea Surface Heights And Derived Variables Reprocessed 1993 Ongo- ing"dataset can be found at https://doi.org/10.48670/moi-00148. The files used to produce Fig. 2,S2 and S3 are available at https://doi.org/10.5281/zenodo.16935712. Seaglider data are available at https://zenodo.org/records/15228200. CTD data are available at https://doi.org/10.17882/95314. The files used to produce Figure S9 are available at https://doi.org/10.5281/zenodo.16935712. Data of BGC-Argo floats WMO6903129 and WMO6904235 can be found at https://fleetmonitoring.euro-argo.eu/float/6903129 and https://fleetmonitoring.euro-argo.eu/float/6904235. The GLO-DAPv2.2023 dataset of the The Global Ocean Data Analysis Project (GLODAP) can be found on the main page of the project https://glodap.info/ or can be accessed directly via the ODV-online browser tool https://explore.webodv.awi.de/.

## Code availability

All scripts used for generating the plots in this paper are available from the corresponding author upon request.

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

## Acknowledgements

This study and the research cruise were funded by the EU's Horizon 2020 research and innovation program under grant agreement N821001 (SO-CHIC). S.S and M.d.d.P. thank those part of the collective effort in piloting the glider during the experiment as well as the captain and crew of the RV SA Agulhas II for the deployment of the glider. S.S. is supported by a Wallenberg Academy. Fellowship (WAF 2015.0186) and the Swedish Research Council (VR 2019-04400). M.d.d.P is supported by the European Union's Marie Skłodowska Curie Individual Fellowship under Project ID 101032683. S.M. is supported by the Research Council of Norway (RCN) via the project "I-CRYME: Impact of CRYosphere Melting on Southern Ocean Ecosystems and biogeochemical cycles" (grant number 335512), the Centre of Excellence "iC3: Center for ice, Cryosphere, Carbon and Climate" (grant number 332635) and by the EU for the project "WOBEC: Weddell Sea Observatory and Biodiversity and Ecosystem Change" (grant number 350906). We would like to thank Catherine Schmechtig for calibrating the BGC-Argo floats.

## Author contributions

N.S. and J.-B.S. conceptualized and supervised the study. N.S., M.d.d.P and S.S. performed the curation of the in-situ data. B. D., N.S., M.d.d.P, and S.M. carried the data analysis and figures realization. All authors contributed to interpreting the results, writing and editing of the present paper. J.-B.S. as PI and S.S. as Co-PI of the SO-CHIC program acquired the funding.

## Competing interests

The authors declare no competing interests.
