## [Transparent Peer Review file · Nature Communications]

A Chlorophyll Halo over Maud Rise in the Southern Ocean

Corresponding Author: Mr Bertrand Ducrocq

Version 0:

Reviewer comments:

Reviewer #1

(Remarks to the Author)

Review Report: A Chlorophyll Halo over Maud Rise in the Southern Ocean

This study investigates the phenomenon of a halo-shaped phytoplankton bloom over Maud Rise, a seamount in the Southern Ocean. Although the concept/mechanism shown in this study is already known, the study used extensive observations from satellite data spanning 25 years, in-situ ship-based measurements, Seaglider observations, and Biogeochemical-Argo floats to provide concrete evidence on the physical and biogeochemical drivers that contribute to this recurring bloom. The authors highlight the Taylor cap mechanism, which promotes upwelling of warm, iron-rich deep water, stimulating phytoplankton growth.

The study provides an insight into physical-biogeochemical coupling in the Southern Ocean, particularly the interactions between ocean-topography and phytoplankton blooms. The paper is organized, systematically progressing from background, data collection, findings, discussion, and conclusions. Figures and maps are good, effectively illustrating chlorophyll concentration variations, sea ice patterns, and temperature profiles.

1. Areas for improvement and suggested revisions

1.1 Clarification on the mechanisms of iron Supply

The study infers iron upwelling as a driver of phytoplankton blooms but lacks direct iron measurements or flux estimations. Include references to existing iron measurements in the region or hypothesize about potential iron sources (e.g., sediment resuspension, sea ice/glacial melt).

1.2 Quantification of phytoplankton productivity

The study focuses on Chlorophyll-a concentration but does not estimate net primary productivity (NPP) or carbon fixation rates. Provide a comparative analysis of productivity rates using remote sensing estimates or existing productivity models for the Southern Ocean.

1.3 Discussion on interannual variability

The study states that the chlorophyll halo is a recurring feature, but there is limited discussion on interannual variability and potential climate drivers. Incorporate a more detailed comparison of bloom intensity across different years and discuss the influence of ENSO, Southern Annular Mode, or changing wind patterns on bloom strength.

1.4 Addressing uncertainties & data limitations

There is no dedicated section discussing uncertainties in remote sensing data, in-situ sampling constraints, and potential methodological biases. Acknowledge the limitations of satellite-derived Chl-a estimates, cloud cover interferences, contamination of signals from sea ice reflectance and possible sensor calibration issues.

2. Future research directions

The study concludes with significant findings but lacks recommendations for further research. Suggest future work such as numerical modeling to simulate phytoplankton bloom dynamics, iron tracer experiments to confirm nutrient flux pathways, and zooplankton grazing studies to determine bloom termination mechanisms.

3. Recommendation

The paper contributes well to ocean biophysical research and is suitable for publication with major revisions. Key revisions should focus on adding biogeochemical context (iron supply), interannual variability, and addressing methodological limitations. This paper presents an advancement in understanding regional marine productivity in the Southern Ocean. The use of multi-source data and the focus on mesoscale physical-biological interactions are explained well. The revisions suggested above will further enhance the scientific depth and impact of the manuscript. The authors are encouraged to address these points before final submission.

Reviewer #2

(Remarks to the Author)

This paper provides a comprehensive description of the Maud Rise Bloom. Both the annual change in chlorophyll, sea surface temperature and sea ice cover from satellite data, and also provide in situ observations (from CTD, glider, and float profiles) of the bloom that forms a halo around the north west side of Maud Rise each year. The authors provide comprehensive support for their conclusion that the magnitude and shape of the bloom are driven by increased upwelling of deep water that is most likely rich in iron. This upwelling is in turn a result of the circulation driven by the Taylor column that sits above Maud Rise. The authors place the Maud Rise bloom in the context of the magnitude and location of other Southern Ocean blooms, ranking it among the some of highest biomass recorded by Argo float profiles throughout the Southern Ocean.

I find this work mostly well-supported and interesting. Recurring blooms in the Southern Ocean driven by processes that have not yet been well-characterized are important for our understanding of how changes in temperature and salinity may impact primary production. The Maud Rise bloom appears to be one of these recurring blooms.

However, I have one major issue with the manuscript and that is the lack of discussion of the interannual variability in the Maud Rise bloom. The authors briefly mention that it may be related to the warm water halo (line 88) and discuss the processes that cause the warm water halo (lines 132-146). Given the satellite data included in the paper, it should be very possible to pursue this idea. Does the magnitude of the bloom track the intensity and extent of the warm water halo? Given the arguments that the authors make about the causes of the bloom (the upwelling of warm Weddell Deep Water) it is very odd that they do not pursue this line of inquiry. The strength of the upwelling should have a direct impact on the measured sst. The authors state that the impact of this paper, and how it is different from previous similar studies of the Taylor column and the Maud Rise bloom, is that they are characterizing "the link between the specific ocean dynamics around Maud Rise and the timing and intensity of the phytoplankton bloom developing over it." (Lines 51-53). But the authors do not currently meet this threshold. This needs to be addressed.

Medium-sized issues:

The authors rely heavily on the presence of a Taylor column/cap above Maud Rise. The authors need to define this term. The authors use the terms Taylor cap and Taylor column interchangeably, do they mean the exact same thing? A sentence about the general characteristics of a Taylor column and its impacts on circulation.

The authors also need to characterize the spatial extent and position of the Taylor column, beyond the temperature and salinity sections. Ideally the approximate boundaries of the column would be calculated overlaid as a contour on any of the numerous map figures included in the paper. Given how strongly the circulation above Maud Rise differs from the circulation around Maud Rise, this should be visible in surface current data from satellite altimetry. Alternatively, if the average position/extent of the column has been characterized in another paper, the position of the column from that paper could be used. The Taylor column is such a fundamental part of this work, and it needs to be visualized.

The oceanographic setting of the Maud Rise bloom is difficult to follow. Specifically, the pattern of sea ice retreat, its position relative to the Weddell gyre, and its location relative to the rest of the Weddell Sea.

The authors mention the Maud Rise polynya several times, and then transition to talking about the bloom. They discuss the polynya as a winter phenomenon, but when the winter polynya opens, it also impacts sea ice retreat (sea ice retreats significantly earlier) and therefore primary production. Perhaps a climatology of sea ice concentration of a slightly larger region for each of the months during sea ice retreat in the Supplemental. It was hard to visualize how sea ice retreats through this area.

The authors mention the topography of the Weddell region, but only ever show very close-cropped view of Maud Rise. They also mention that Maud Rise is positioned in the westward flowing limb of the Weddell Gyre, but there is no figure showing the Weddell Gyre. Authors need a figure showing the regional bathymetry and average position of the relevant part of the Weddell Gyre. The fact that Maud Rise falls in the path of the gyre seems important to the presence of the Taylor column.

Only people who work specifically in the Weddell region will have an image of this in their minds. More details on the oceanographic setting of Maud Rise and the associated bloom, related to the above comments, need to be added to the text as well. Lines 45-53 region.

Figure 3 is very important to the arguments being made in the paper, and extremely difficult to make sense of. There has got to be an easier way to label figure 3. Use different symbols and then put those symbols above the lines on the sections. Write above the sections when the float is above the rise and when it isn't. Just something to make this more digestible. It is too hard to digest this figure and even when I thought I had the orientation of everything correct, I wasn't sure. Considering the importance of this figure to the paper it needs clearer annotation. Figure C1 is much clearer and easier to get oriented to.

The authors need to provide the data for the integrated chlorophyll values that they calculated from Argo float profiles. This is the data on which the importance of the Maud Rise bloom is based.

Small issues:

In the caption for Figure 1 the Authors mention isopycnals drawn on the sections, but I cannot find the details of how density was calculated. This needs to be a sentence in the methods.

For figure 2e, the caption mentions sea ice contours, but all I see are bathymetry contours.

Version 1:

Reviewer comments:

Reviewer #2

(Remarks to the Author)

I added several figures to my review and have included my review as an attachment as the figures could not be copied here and the review does not make much sense without them.

(Remarks on code availability)

An alternate way to address my comments on the Argo float data availability for the data presented in figure A4 would be to provide the code used to make the corrections to the chlorophyll data.

Version 2:

Reviewer comments:

Reviewer #2

(Remarks to the Author)

Review Round 3

This paper describes the phytoplankton bloom that forms around Maud Rise in the eastern Weddell Sea. They link phytoplankton abundance to upwelling of warm/salty deep water layer due to the mesoscale dynamics on the edges of the Taylor column. They show the thinning of the winter water layer and the intrusion of the warm/salty deep water with glider, argo float, and ctd data. In addition to the previous addition showing the connection between changes in chlorophyll a concentration and changes in SST they have now added a figure showing the connection between changes in SST with changes in SSH.

The authors have addressed all of my previous comments to my satisfaction. I have one more general comment, and a few specific comments shown below. I believe these are all very easily handled with very minor changes to the manuscript.

General comment:

In the section where you talk about the relevance of the Maud Rise Bloom and use the Argo float data to highlight how productive it is compared to other blooms in the Southern Ocean, you highlight that Maud Rise is responsible for 11 of the 39 profiles you find in the Argo record with integrated Chl over 100 mg m⁻³. Unless I am misunderstanding something this feels like an overstatement. Not that it isn't true, but what I understand from your paper is that these profiles are from the float that you deployed into these productive waters, while the other high chl profiles were encountered by chance. If I have misunderstood this, please make it clearer in your paper that this was not the float shown in Figure 1b that was specifically deployed into the bloom. If I have this correct, then it is inappropriate to play this up as you have introduced a sampling bias into the integrated chl data set by deploying a float into a known area of high chl and then comparing it to the frequency of other regions of high chl in the rest of the dataset. This needs to be made clear in your text. Either by clarifying that this data is from a different float or by acknowledging the sampling bias and addressing it directly in the text.

Specific comments:

Figure 1 caption: You say that (d) and (e) show a longitudinal transect when it is clearly latitudinal, judging from the x-axis which shows it going from east to west. If you're looking to say this in a single word zonal or latitudinal work.

Figure 4 caption: Add the definition that you use for MLD back to the caption. You had it there in a previous version, please put it back.

Figure 4: You don't include the symbols above the sections that correspond to the symbols in the glider track maps. You had these in the last version. Please add them back.

Line 163: do you mean GLODAP instead of GLOBDATA?

Line 168 and Line 171: you left the "He" out of $\delta^3\text{He}$

Figure A4: Adjust the colorscale in Figure A4a to fit the range of the data which appear to go from -1 to 1 rather than -5 to 5.

Figure A4: Add hatching to indicate area where p-value was less than 0.05 on both A4 figures

Casey Schine

REVIEWER COMMENTS

Reviewer #1 (Remarks to the Author):

Review Report: A Chlorophyll Halo over Maud Rise in the Southern Ocean

We thank the reviewer for their feedback to improve our manuscript, which we have revised accordingly. We appreciate the time and energy dedicated to evaluating this manuscript, and we recognise how the constructive and critical feedback has helped us improve the quality and clarity of the work presented. Below are the original comments from the reviewer in black and our corresponding responses in green. Text from the manuscript is shown in green and italic.

This study investigates the phenomenon of a halo-shaped phytoplankton bloom over Maud Rise, a seamount in the Southern Ocean. Although the concept/mechanism shown in this study is already known, the study used extensive observations from satellite data spanning 25 years, in-situ ship-based measurements, Seaglider observations, and Biogeochemical-Argo floats to provide concrete evidence on the physical and biogeochemical drivers that contribute to this recurring bloom. The authors highlight the Taylor cap mechanism, which promotes upwelling of warm, iron-rich deep water, stimulating phytoplankton growth.

The study provides an insight into physical-biogeochemical coupling in the Southern Ocean, particularly the interactions between ocean-topography and phytoplankton blooms. The paper is organized, systematically progressing from background, data collection, findings, discussion, and conclusions. Figures and maps are good, effectively illustrating chlorophyll concentration variations, sea ice patterns, and temperature profiles.

We thank the reviewer for these positive comments.

1. Areas for improvement and suggested revisions 1.1 Clarification on the mechanisms of iron Supply

The study infers iron upwelling as a driver of phytoplankton blooms but lacks direct iron measurements or flux estimations. Include references to existing iron measurements in the region or hypothesize about potential iron sources (e.g., sediment resuspension, sea ice/glacial melt).

Our analysis hypothesized that the observed bloom over Maud Rise is associated with iron upwelling supply from deeper waters. Such a mechanism assumes that a deep iron source provides the necessary iron that is needed to allow for such high biomass build-up. As discussed in Moreau et al. (2023) or in Ardyna et al. (2019), hydrothermal iron can be transported thousands of kilometers from its origin and be a key driver of surface primary production. In the sector of Maud Rise, meridional section of Helium isotopic ratio ($\delta^3\text{He}$), which is a tracer of primordial helium originating from the Earth's mantle, commonly used to detect plumes downstream of hydrothermal vents, shows elevated ratios directly below the surface layer (see Figure 3b of Moreau et al., 2023). This clearly indicates that water directly below the surface layer in our region of interest does have influence of hydrothermal vents that are known to be loaded in iron. We have clarified these aspects in the new version of the manuscript. Also, based on the recommendation of the reviewer regarding the potential role of sea ice, we added the following sentence stating that some of the iron supply could come from the sea ice melt (lines 156-157):

“It is also probable that some part of the iron suspended in the upper sunlit layers originates from sea ice melting [6].”

However, we do not think that sediment resuspension could play a role in this deep-sea environment, the shallowest parts of Maud Rise being deeper than 1,000 m deep.

References:

Ardyna, M., Lacour, L., Sergi, S. et al. Hydrothermal vents trigger massive phytoplankton blooms in the Southern Ocean. Nat Commun 10, 2451 (2019). <https://doi.org/10.1038/s41467-019-09973-6>

Moreau, S., Hattermann, T., de Steur, L. et al. Wind-driven upwelling of iron sustains dense blooms and food webs in the eastern Weddell Gyre. Nat Commun 14, 1303 (2023). <https://doi.org/10.1038/s41467-023-36992-1>

D. Lannuzel, M. Vancoppenolle, P. van der Merwe, J. de Jong, K.M. Meiners, M. Grotti, J. Nishioka, V. Schoemann; Iron in sea ice: Review and new insights. Elementa: Science of the Anthropocene 1 January 2016; 4 000130. <https://doi.org/10.12952/journal.elementa.000130>

1.2 Quantification of phytoplankton productivity

The study focuses on Chlorophyll-a concentration but does not estimate net primary productivity (NPP) or carbon fixation rates. Provide a comparative analysis of productivity rates using remote sensing estimates or existing productivity models for the Southern Ocean.

The reviewer is right and the study of Net primary productivity (NPP) could bring interesting results to the present study, such as using the remote-sensing based VGPM or CbPM models of Berenfeld et al. (1997, 2005). However, there are also inherent uncertainties associated with these models and the carbon- and chlorophyll-based models typically result in opposing trends as shown by Pinkerton et al. (2021). In addition, we believe that presenting NPP data will not improve, in this context, our understanding of the formation mechanism of the annular phytoplankton blooms. Given the amount of data we already present in the article, for e.g., with more than 2 decades of information about satellite-derived ocean colour, we do not wish to increase overly the size of the manuscript by adding the study of remotely-sensed ocean NPP, also to remain within the standard size of articles published in Nature Communication. We believe that the study of the phytoplankton biomass (inferred through satellites but also in situ with CTD casts, floats and gliders) is relevant to make a strong link with ocean dynamics as we present in this study. We leave it to the editor and the reviewer to decide whether they wish to include such an analysis of the satellite-derived NPP to the article.

References :

Behrenfeld et al. ; A consumer's guide to phytoplankton primary productivity models ; 22 December 2003 ; <https://doi.org/10.4319/lo.1997.42.7.1479>

Behrenfeld et al. ; Carbon-based ocean productivity and phytoplankton physiology from space ; 25 January 2005 ; <https://doi.org/10.1029/2004GB002299>

Pinkerton et al ; Evidence for the impact of climate change on primary producers in the southern ocean ; 17 May 2021 ; <https://doi.org/10.3389/fevo.2021.592027>

1.3 Discussion on interannual variability

The study states that the chlorophyll halo is a recurring feature, but there is limited discussion on interannual variability and potential climate drivers. Incorporate a more detailed comparison of bloom intensity across different years and discuss the influence of ENSO, Southern Annular Mode, or changing wind patterns on bloom strength.

The interannual variability of the phytoplankton bloom over the area of Maud Rise is shown in figure 2a and appendix figure A1. Figure 2a suggests that there is a strong interannual variability in phytoplankton biomass while appendix figure A1 suggests a moderate spatial variability of the summer phytoplankton bloom, although with recurring high biomass along the flanks of Maud Rise.

To respond to the reviewer's comment, in the new version of the manuscript, we added figure 3 and appendix figures A2 showing the interannual variability in the warm water halo around Maud Rise (figure A2) and the significant correlation between the warm water halo and the annular phytoplankton bloom (figure 3). In addition, we added a paragraph discussing this aforementioned link to the end of the section "A recurring halo-shaped phytoplankton bloom over Maud Rise" (lines 101-110):

"The strength of the phytoplankton bloom show strong interannual variability but moderate spatial variability, with recurring high biomass along the flanks of Maud Rise (Figures 2a and A1). The warm water pattern generally matches the phytoplankton bloom pattern (Figures A1 and A2). The correlation map (Figure 3) between the monthly time series of SST and Chl-a concentration indicates a significantly positive correlation on the flanks of the Maud Rise (nearly $r = 0.7$ on the eastern side directly facing the incoming flow of the Weddell gyre). Therefore, we suggest that the interannual variability of the chlorophyll halo can, to a great extent, be explained by the interannual variability of the warm water halo [35]. In addition, we suggest that anomalies in easterly winds may be responsible for enhanced upwelling in this part of the Southern Ocean and, therefore, lead to higher phytoplankton biomass as shown by [10] in the region directly Southeast of Maud Rise."

We also do not believe that ENSO or SAM play a significant role on phytoplankton dynamics around Maud Rise, as there are years when strong positive and negative ENSO or SAM anomalies were detected but with no discernable anomalies in the intensity of the annular phytoplankton blooms. Although phytoplankton dynamics have been shown to be related to the ENSO or SAM index in western parts of Antarctica like the Antarctic Peninsula (e.g., Vernet et al., 2008; Moreau et al., 2015), which the reviewer is perhaps referring to, such links do not seem to exist in eastern parts of Antarctica as suggested by Moreau et al. (2023) who carried these analyses in the region directly Southeast of Maud Rise.

As the reviewer suggested, however, changing wind patterns may indeed have an impact on phytoplankton blooms as also suggested by Moreau et al. 2023 who investigated the drivers of a recurring phytoplankton bloom Southeast of Maud Rise. The authors found that this anomalously productive bloom was linked to negative anomalies in easterly winds which, in the Southern Ocean, indicate enhanced upwelling. A similar but unpublished relationship was observed between phytoplankton biomass and easterly winds anomalies at Maud Rise (L. de Steur, pers. comm.) Therefore, following the reviewer's comment, we added the following sentences to the new version of the manuscript (lines 103-110) :

"The correlation map (Figure 3) between the monthly time series of SST and Chl-a concentration indicates a significantly positive correlation on the flanks of the Maud Rise (nearly $r=0.7$ on the eastern side directly facing the incoming flow of the Weddell gyre). Therefore, we suggest that the interannual variability of the chlorophyll halo can, to a great extent, be explained by the interannual variability of the warm water halo [35]. In addition, we suggest that anomalies in easterly winds may be responsible for enhanced upwelling in this part of the Southern Ocean and, therefore, lead to higher phytoplankton biomass as shown by [10] in the region directly Southeast of Maud Rise."

References :

Vernet et al. Primary production within the sea-ice zone west of the Antarctic Peninsula : I-sea ice, summer mixed layer, and irradiance ; September 2008 ; <https://doi.org/10.1016/j.dsr2.2008.05.021>

Moreau et al. Climate change enhances primary production in the western Antarctic Peninsula ; 28 January 2015 ; <https://doi.org/10.1111/qcb.12878>

1.4 Addressing uncertainties & data limitations

There is no dedicated section discussing uncertainties in remote sensing data, in-situ sampling constraints, and potential methodological biases. Acknowledge the limitations of satellite-derived Chl-a estimates, cloud cover interferences, contamination of signals from sea ice reflectance and possible sensor calibration issues.

We have now added the following to more fully describe the satellite based observations methods that are found in Methods (lines 207-211).

The satellite-derived Chl-a concentration is evaluated by observations in the visible domain of the ocean surface implying that cloud cover and sea ice hinder the retrieval of upper ocean Chl-a concentrations [41]. The sea surface temperature and sea ice concentration estimates used in this study rely on the sea surface reflectance in the near infrared domain which is little affected by clouds but is subject to sea ice reflectance interferences and possible sensor calibration issues.

We were not sure what further details the reviewer is requesting for the in situ observations. The in situ glider and ship-based observations are described in detail in the Methods section. Aspects such as measurement quality control, calibrations against other measurements and CTDs, corrections (including correction for non-photochemical quenching) are provided.

2. Future research directions

The study concludes with significant findings but lacks recommendations for further research. Suggest future work such as numerical modeling to simulate phytoplankton bloom dynamics, iron tracer experiments to confirm nutrient flux pathways, and zooplankton grazing studies to determine bloom termination mechanisms.

The following sentences were added at the end of conclusion serving the purpose of suggesting further research (lines 191-199) :

As such, phytoplankton blooms over Maud Rise constitute a good case study of how marine ecosystems are shaped by mesoscale ocean dynamics. More investigations are needed concerning the iron supply, the role of grazers on bloom dynamics as well as the impact of the local state of the atmosphere on the strength of the blooms. Although trace metal nutrient measurements are complex to make, further trace metal measurements in open-ocean polynya regions, especially in the presence of shallow topography, would help elucidate the nutrient supply mechanisms between the deep ocean and surface layers. Understanding such dynamics explained in the present study would benefit from a computer simulations-based investigation including biogeochemical components. This could help disentangle the bio-physical processes occurring in these complex polynya systems.

3. Recommendation

The paper contributes well to ocean biophysical research and is suitable for publication with major revisions. Key revisions should focus on adding biogeochemical context (iron supply), interannual variability, and addressing methodological limitations. This paper presents an advancement in understanding regional marine productivity in the Southern Ocean. The use of multi-source data and the focus on mesoscale physical-biological interactions are explained well. The revisions suggested above will further enhance the scientific depth and impact of the manuscript. The authors are encouraged to address these points before final submission.

We thank the reviewer for their constructive comments that we believe improved the quality of the manuscript.

Reviewer #2 (Remarks to the Author):

This paper provides a comprehensive description of the Maud Rise Bloom. Both the annual change in chlorophyll, sea surface temperature and sea ice cover from satellite data, and also provide in situ observations (from CTD, glider, and float profiles) of the bloom that forms a halo around the north west side of Maud Rise each year. The authors provide comprehensive support for their conclusion that the magnitude and shape of the bloom are driven by increased upwelling of deep water that is most likely rich in iron. This upwelling is in turn a result of the circulation driven by the Taylor column that sits above Maud Rise. The authors place the Maud Rise bloom in the context of the magnitude and location of other Southern Ocean blooms, ranking it among the some of highest biomass recorded by Argo float profiles throughout the Southern Ocean.

I find this work mostly well-supported and interesting. Recurring blooms in the Southern Ocean driven by processes that have not yet been well-characterized are important for our understanding of how changes in temperature and salinity may impact primary production. The Maud Rise bloom appears to be one of these recurring blooms.

We appreciate the time and energy dedicated to evaluating this manuscript, and we recognise how the constructive and critical feedback has helped us improve the quality and clarity of the work presented.

However, I have one major issue with the manuscript and that is the lack of discussion of the interannual variability in the Maud Rise bloom. The authors briefly mention that it may be related to the warm water halo (line 88) and discuss the processes that cause the warm water halo (lines 132-146). Given the satellite data included in the paper, it should be very possible to pursue this idea. Does the magnitude of the bloom track the intensity and extent of the warm water halo? Given the arguments that the authors make about the causes of the bloom (the upwelling of warm Weddell Deep Water) it is very odd that they do not pursue this line of inquiry. The strength of the upwelling should have a direct impact on the measured SST. The authors state that the impact of this paper, and how it is different from previous similar studies of the Taylor column and the Maud Rise bloom, is that they are characterizing “the link between the specific ocean dynamics around Maud Rise and the timing and intensity of the phytoplankton bloom developing over it.” (Lines 51-53). But the authors do not currently meet this threshold. This needs to be addressed.

Thank you for bringing this to our attention. By being in the details of the analysis we may have missed this crucial part of the study, and your comment has considerably helped us to bring this SST-chl relationship more prominently into the study. As you will see with the figure 3 below, there is indeed a remarkably strong relationship between SST and chl and therefore between the ocean dynamics and the phytoplankton bloom over Maud Rise. We now bring these updated findings into the manuscript. Thank you again for this insightful point.

In more details, we added the two figures below to the main manuscript and to the section A of the appendix as well as the following paragraph at the end of the section 3.1 “A recurring halo-shaped phytoplankton bloom over Maud Rise” (lines 101 - 110) :

The strength of the phytoplankton bloom show strong interannual variability but moderate spatial variability, with recurring high biomass along the flanks of Maud Rise (Figures 2a and A1). The warm water pattern generally matches the phytoplankton bloom pattern (Figures A1 and A2). The correlation map (Figure 3) between the monthly time series of SST and Chl-a concentration indicates a significantly positive correlation on the flanks of the Maud Rise (nearly $r = 0.7$ on the eastern side directly facing the incoming flow of the Weddell gyre). Therefore, we suggest that the interannual variability of the chlorophyll halo can, to a great extent, be explained by the interannual variability of the warm water halo [35]. In addition, we suggest that anomalies in easterly winds may be responsible for enhanced upwelling in this part of the Southern Ocean and, therefore, lead to higher phytoplankton biomass as shown by [10] in the region directly Southeast of Maud Rise.

Figure A2 : Annual climatologies of the warm water halo. Spatial maps of SST concentration obtained by averaging satellite data over December for each year. Isobaths, plotted as black solid contours, are drawn every 500 meters. The 3500 meter isobath is drawn in white.

Figure 3 : Correlation map between the SST and the Chl-a. The two time series used to compute the correlation map are the monthly time series used to plot the Figures A1 and A2. Isobaths, plotted as black solid contours, are drawn every 500 meters. The 3500 meter isobath is drawn in white.

Medium-sized issues:

The authors rely heavily on the presence of a Taylor column/cap above Maud Rise. The authors need to define this term. The authors use the terms Taylor cap and Taylor column interchangeably, do they mean the exact same thing? A sentence about the general characteristics of a Taylor column and its impacts on circulation.

The following paragraph was added to the introduction (lines 45-49) :

According to the Taylor-Proudman theorem, the water column above a seamount is stiffened by Earth's rotation which effectively traps the water masses within the column and blocks any incoming flow as if the seamount extended up to the surface [25, 31,32]. This concept known as Taylor column is a simple theoretical model, but describes well the Maud Rise ocean dynamics. The hydrography of the stagnant water masses overlying the seamount can significantly differ from the incoming flow [16].

The term "Taylor cap" has been replaced by "Taylor column" throughout the manuscript. We thank the review for bringing this detail up to our attention.

The authors also need to characterize the spatial extent and position of the Taylor column, beyond the temperature and salinity sections. Ideally the approximate boundaries of the column would be calculated overlaid as a contour on any of the numerous map

figures included in the paper. Given how strongly the circulation above Maud Rise differs from the circulation around Maud Rise, this should be visible in surface current data from satellite altimetry. Alternatively, if the average position/extent of the column has been characterized in another paper, the position of the column from that paper could be used. The Taylor column is such a fundamental part of this work, and it needs to be visualized.

Thank you for raising this point. As the reviewer suggests, the Taylor column should be visible in surface current data, and indeed, a doming is visible in the climatology of the sea surface height (SSH), which we added to figure 2g in the new version of the manuscript (see below). However, even though the theory of the Taylor column is based on the rotating shallow water hypothesis, at Maud Rise, the Taylor column and its SSH signal are attenuated by the stratification (isopycnals doming) (Chapman et Haidgovel, 1991). Previous simulations-based work in the literature have shown that, under a strong flow, the Taylor column can be swept off from the seamount (Chapman et Haidgovel, 1991). The warm water halo is well hugged by the 3500 meter isobath and we decide to use it as an approximate characterization of the Taylor column's extent. For clarity, the 3500 meter isobath has been added to multiple maps across the manuscript (Figure 1b, 2d.g, 3, 4a-b, A1, A2, A3, and C1a-b).

Reference :

Chapman et Haidgovel, 1991, Formation of Taylor caps over a tall isolated seamount in a stratified ocean.
<https://doi.org/10.1080/03091929208228084>

Figure 2 : Characterization of the halo structure through climatologies of the satellite observations. Seasonal cycle of (a) Chl-a concentration, (b) Sea Surface Temperature (SST) and (c) fraction of sea ice cover over the area of the Maud Rise (5W-10E, 62S-68S). For each of those three variables, each year from 1998 to 2023 is plotted as a solid black line except the cruise year 2021-2022 represented as a solid blue line. The solid red line is the average over the period 1998-2023 and the red filled area represents the variation by one standard deviation. (d) January climatology of the satellite Chl-a concentration between 1998 to 2023. (e) Climatology of sea ice concentration over the period mid-November to mid-December between 1998 and 2022. (f) December climatology for SST averaged between 1998 to 2022. (g) January climatology for sea surface height (SSH) averaged between 1998 to 2023. The thick blue/white line in (d), (e), (f), (g) is the 3500 meter isobath.

The oceanographic setting of the Maud Rise bloom is difficult to follow. Specifically, the pattern of sea ice retreat, its position relative to the Weddell gyre, and its location relative to the rest of the Weddell Sea. The authors mention the Maud Rise polynya several times, and then transition to talking about the bloom. They discuss the polynya as a winter phenomenon, but when the winter polynya opens, it also impacts sea ice retreat (sea ice retreats significantly earlier) and therefore primary production. Perhaps a climatology of sea ice concentration of a slightly larger region for each of the months during sea ice retreat in the Supplemental. It was hard to visualize how sea ice retreats through this area.

We thank the reviewer for bringing this up to our attention. Based on the reviewer's comment, a climatology of sea ice concentration and the major oceanographic currents of a slightly larger region encompassing the Weddell Gyre were plotted and added to the appendix (Figure A3). We hope that this provides the reader with a broader context of the connections between the Maud Rise, sea ice concentration and the phytoplankton blooms described in this study.

Sea Ice Concentration Climatology

Figure A3 Sea Ice Concentration Climatology of the Austral Summer : spatial maps of sea ice concentration obtained by averaging satellite data over 4 different periods from 1998 to 2022. The 3800 meter isobath is drawn as a solid black line. ACC, WDW and CDW stand for Antarctic Circumpolar Current, Weddell Deep Water and Cold Deep Water respectively.

The authors mention the topography of the Weddell region, but only ever show a very close-cropped view of Maud Rise. They also mention that Maud Rise is positioned in the westward flowing limb of the Weddell Gyre, but there is no figure showing the Weddell Gyre. Authors need a figure showing the regional bathymetry and average position of the relevant part of the Weddell Gyre. The fact that Maud Rise falls in the path of the gyre seems important to the presence of the Taylor column. Only people who work specifically in the Weddell region will have an image of this in their minds.

More details on the oceanographic setting of Maud Rise and the associated bloom, related to the above comments, need to be added to the text as well. Lines 45-53 region.

In response to the reviewer's comment, and as described in our response to the previous comment, arrows representing the major oceanographic currents in the Weddell Gyre were added to the new figure A3 showing the development of the Maud Rise Polynya, thus giving a better insight of the general circulation in the Weddell Gyre and around Maud Rise.

Figure 3 is very important to the arguments being made in the paper, and extremely difficult to make sense of. There has got to be an easier way to label figure 3. Use different symbols and then put those symbols above the lines on the sections. Write above the sections when the float is above the rise and when it isn't. Just something to make this more digestible. It is too hard to digest this figure and even when I thought I had the orientation of everything correct, I wasn't sure. Considering the importance of this figure to the paper it needs clearer annotation. Figure C1 is much clearer and easier to get oriented to.

We thank the reviewer for this suggestion that improved the readability of this Figure. In response, we added different symbols where the seaglider was deployed and above the corresponding lines, to better link the sections with the maps (see the new version of the figure 3).

Figure 3 High resolution probing of the halo by seaglider during the austral summer 2021-2022 : (a) Maximum temperature measured between 0 and 200 meters by the glider survey. (b) Glider observed integrated Chl-a concentration between 0 and 200

meters. For both (a) and (b), isobaths are drawn every 200 meters and the 3500 meter isobath is shown in blue. Blue lines in the glider sections of (c) Chl-a, (d) temperature, and (e) salinity correspond to checkpoints marked by different blue-bounded white symbols in (a) and (b). Isopycnals are drawn every 0.1 kg/m^3 between 1023.0 and 1030.0 kg/m^3 . The white star and hexagon in (a) and (b) marks where the time series shown in (c), (d) and (e) begin and end. The thick black line in (c), (d), (e) is the mixed layer depth based on the definition used by Xing et al, 2012.

The authors need to provide the data for the integrated chlorophyll values that they calculated from Argo float profiles. This is the data on which the importance of the Maud Rise bloom is based.

The link to the data used for assessing the importance of the Maud Rise bloom was mistakenly put in the Acknowledgement section. The Data availability section now reads (lines 202-217) :

The satellite observations used in this study are based on the "Global Ocean Colour Plankton and Reflectances MY L3 daily observations" dataset with daily and $4 \text{ km} \times 4 \text{ km}$ resolutions, the "Global Ocean Gridded L4 Sea Surface Heights And Derived Variables Reprocessed 1993 Ongoing" dataset with daily and $0.125^\circ \times 0.125^\circ$ resolutions, and the "Global Ocean OSTIA Sea Surface Temperature and Sea Ice Reprocessed" dataset with daily and $0.05^\circ \times 0.05^\circ$ resolutions. The three datasets extend from 1998 to 2022 (SIC, SST) or 2023 (CHL-a, SSH). The satellite-derived Chl-a concentration is evaluated by observations in the visible domain of the ocean surface implying that cloud cover and sea ice hinder the retrieval of upper ocean Chl-a concentrations [42]. The sea surface temperature and sea ice concentration estimates used in this study rely on the sea surface reflectance in the near infrared domain which is little affected by clouds but is subject to sea ice reflectance interferences and possible sensor calibration issues. Based on the satellite-derived sea ice concentration, we define the fraction of the Maud Rise area covered by sea ice as the number of pixels with sea ice concentration above 15% in the domain (5°W - 10°E , 62°S - 68°S) divided by the total number of pixels in the same domain. Chl-a concentration derived from satellite imagery matches the measurements in situ (Figures 1 b and 1 d). The correlation map (Figure 3) was computed first by calculating the yearly time series of Chl-a concentration and sea surface temperature (Figures A1 and A2) then computing the Pearson correlation coefficient map for these two time series.

Small issues:

In the caption for Figure 1 the Authors mention isopycnals drawn on the sections, but I cannot find the details of how density was calculated. This needs to be a sentence in the methods.

The following sentence was added at the end of the subsection "In-situ Observations" of the "Methods" section (lines 242-244) :

Isopycnals were calculated based on the Thermodynamic Equation of Seawater - 2010 via the gsw-python library that calculates in-situ density of seawater from absolute salinity and in-situ temperature.

For figure 2e, the caption mentions sea ice contours, but all I see are bathymetry contours.

We thank the reviewer for noticing this mistake. We decided not to draw sea ice contours as we thought it does not significantly help to understand the figure 2e. Therefore, we corrected the caption to no longer mention sea ice contours. We also decided to remove the sea surface temperature contours for the same reasons in figure 2f.

Review Round 2

This paper describes the phytoplankton bloom that forms around Maud Rise in the eastern Weddell Sea. They link phytoplankton abundance to upwelling of warm/salty deep water layer due to the mesoscale dynamics on the edges of the Taylor column. They show the thinning of the winter water layer and the intrusion of the warm/salty deep water with glider, argo float, and ctd data. They have also now linked changes in chlorophyll a concentration with changes in SST indicative of the upwelling of the warm deep water.

I very much want to see this paper published. I think the findings are very interesting. However, I still think there are issues that need to be addressed. I've detailed them in three sections below.

We thank the reviewer for the positive and constructive comments about our paper. We respond below to all these new comments. We believe that they have significantly improved the quality of the manuscript and we hope that both the editor and the reviewer will find this new version of the paper acceptable for publication in Nature Communication.

Stats reporting

The analysis of the relationship between chl and sst in Figure 3 does a good job of addressing my comments about interannual variability from the last round of reviews. However, there needs to be more information given on the relationship identified beyond just the correlation coefficient. I think it would be appropriate to report the p-value and the slope of the relationship. I don't expect that the p-value would tell a very different story than the correlation coefficient already presented, but it should be included. The slope gives us additional information on the magnitude of the relationship not just the strength of it. How much does the chl change with increasing or decreasing SST? I would recommend adding these to the supplemental and referencing them briefly in the text.

We thank the reviewer for suggesting that we include the p-value and slope. Statistically significant (p -value < 0.05) regions were hatched on the correlation maps. The slopes of the SSH-SST and Chl-a - SST correlation maps were added as a single figure in the appendix (Figure A4).

Figure 3: **Correlation maps of satellite variables.** Correlation maps between (a) SST and Chl-*a* concentration and (b) between SST and SSH. Isobaths, plotted as black solid contours, are drawn every 500 m. The 3500 m isobath is drawn in blue. Areas with a p -value under 0.05 are hatched. See Figure A4a for more information on the slope of the regression.

Figure A4: **Slope of the correlation.** a) Slope of the Chl-*a* -SST correlation b) Slope of the SSH-SST correlation. Isobaths, plotted as black solid contours, are drawn every 500 meters. The 3500 meter isobath is drawn in Blue.

I also feel like you are leaving something on the table by not taking your analysis further by now looking at the relationship between SSH and SST, or chl and SSH. It would be a very powerful connection if there is in fact a significant correlation between either of these variables and SSH. Then you tie the strength of the upwelling (and the bloom) to the strength of the Taylor column. I'm not going to insist on it, but it would be an excellent addition to your narrative. And seemingly with all of the data you have put together and the code you have presumably already developed, it should be a relatively easy analysis.

Although no interesting correlation was found between SSH and Chl-*a* concentration, a positive correlation between SSH and SST was found as explained in a new paragraph (lines 112-115) added at the end of the section titled "A recurring halo-shaped phytoplankton bloom over Maud Rise":

*A significant correlation between the SST and SSH yearly timeseries was also observed, with values reaching up to $r \approx 0.6$ on the Northeastern flank that directly faces the incoming westward flow (Figure 3b). These two correlations, between SST and SSH between and SST and the Chl-*a* concentration, point towards the pivotal role of ocean-topography interaction in the formation of chlorophyll halos as discussed in the next section.*

The correlation map was added in the main text as Figure 3b).

Data availability

You need to make the data that you used in your analyses available publicly. In my last review I asked for the publication of the Argo data that you used to make the maps in Figure A4, and you responded with a paragraph of text about satellite data. Clearly this was a mistake, however the data still have not been published.

We apologize for this oversight in our previous response to the reviewer's comment. As described below, we now provide all the BGC-Argo float data used in this paper.

I mocked up a version of the same figure as A4 from some float profiles that I have for the Pacific sector of the SO that shows all of the profiles with integrated chl (down to 200 meters) above 100 mg m⁻². There are a lot more than what you show in your figure A4. Out of 1227 profiles I have 146 with integrated chl above 100 mg m⁻² (11%) versus the 39 out of 8560 that you found (0.4%).

It's not that I don't believe your results, it's that I don't think you provide enough information for me to get the same answer. You hang the importance of the Maud Rise bloom on the magnitude relative to other blooms in the Southern Ocean, at a minimum you need to publish these data, and you should also add more detail on the chl fluorescence corrections used. If the version of the Argo chl data that you used is a publicly available product, then link to that. But the link to the Argo website and the original (presumably uncorrected) float data results in the answer that I show in the figure below. I need to understand why it's different and how to get the answer that you get, and then you need to provide the data.

We apologize for not describing this more clearly in the previous versions of the article. We used BGC-Argo floats data from the SOCCOM program to produce these results. For each float, both an uncorrected Chlorophyll-a variable (named "Chl_a") and a corrected Chlorophyll-a variable (named "Chl_a_corr") are provided. We have consistently used the corrected Chlorophyll-a variable ("Chl_a_corr") which is consistently smaller than the uncorrected Chlorophyll-a variable ("Chl_a"). We wonder if the reviewer used the uncorrected Chlorophyll-a variable to produce the results mentioned above.

In detail, the "Chl_a_corr" variable provided with SOCCOM BGC-Argo floats contains corrected chlorophyll-a values, where several post-processing steps have been applied to improve accuracy, including dark correction to minimize sensor baseline drift, NPQ (non-photochemical quenching) correction to account for the underestimation of fluorescence in sunlit surface waters, and quality control steps in accordance with BGC-Argo standard protocols.

However, to make sure that there are no further doubts on the type of data we are using, and as the reviewer suggested, in the new version of the manuscript, we now provide a link to the BGC-Argo floats data used as well as the code used to produce the integrated Chl-a biomass presented in the manuscript (Figure B1). We hope that the reviewer will find this approach acceptable.

Line 301 : *The code and files used to produce Figure B1 are available at the following location: <https://doi.org/10.5281/zenodo.16935712>.*

I didn't ask for you to publish the satellite data in my last review, because you were only using them for descriptive purposes, but now you are running statistical analyses and you need to put the data that you used in a public repository. This includes the data for the images in figures A1 and A2 (the December averages for chl and sst for each year) as well as the data used to calculate the sea ice and SSH climatologies in Figure 2. Linking to the website that provides the daily satellite images is not enough. This would require someone to download daily images for the relevant time period for each year (~775 images for each of 4 different satellite products) and then create averages for each year, just to evaluate your findings. You do state that code to make the figures is available upon request. This is not the current standard for publication. The data need to be publicly available. You should probably include the glider data too.

The data used for plotting figures 2e, 2g, A1 and A2 is now publicly available at <https://doi.org/10.5281/zenodo.16935712>. This is mentioned in the text, lines 248-249.

We added the following link for the Seaglider data (lines 264-265):
Seaglider data are available at <https://zenodo.org/records/15228200>.

Referencing the float numbers and website for the float data that you used seems fine as those are curated in a way that makes them very easy to access.

Deep iron concentrations

In reading through the comments from Reviewer 1, my attention was drawn to the source of the iron being delivered from the deep water, and I went down a bit of a rabbit hole and found several issues. As reviewer 1 identified, you do not really resolve the issue of whether the water mass being upwelled does in fact have enough Fe to generate the observed bloom. And the changes that you made in response to their comment, do not really address this issue.

This needs to be addressed but I think it is a simple fix. In your response to reviewer 1 you mention hydrothermal Fe and the elevated ^3He found in the eastern Weddell, but you never mention findings on this in the paper and only reference papers that say this without ever mentioning the word hydrothermal. I have read the Moreau paper that you reference previously, but in looking back at it, one of the transects that measures ^3He , practically runs over Maud Rise. You can directly address the origin of the Fe by plotting a section of the ^3He from this transect and then do a water mass analysis that shows that the elevated ^3He is in fact associated with the WDW water mass. This was the method used to link phytoplankton blooms near the SWIR to hydrothermal vents upstream in the Ardyna paper that you also reference, and would satisfy me that we could reasonably expect elevated Fe concentrations in the WDW water mass.

To satisfy my own curiosity and to not be that reviewer that sends you down a horribly unproductive blind alley, I took a look at the ^3He data for the transect in question (shown below; bottom left shows the map with the ^3He transect shown in the top right outlined in red, top left shows a map of the maximum ^3He value for each profile in color, and bottom right is a TS plot with ^3He shown in color). You can actually see in the section plot where the transect passes over the shoulder of Maud Rise at 65S and there is clearly an elevated ^3He anomaly there. This should be a pretty straightforward connection to make simply by referencing the T/S range of your warm/salty water mass and mapping it onto the T/S space for the transect.

Thanks for the careful analysis to help us improve our argument that iron is upwelled from WDW. Following the reviewer's suggestion we added the following paragraph to the new version of the manuscript, lines 163-176, as well as 3 figures in the appendix, Figure C1,C2 and C3 :

We verified this iron supply hypothesis by using the GLOPDATA database and its $\delta^3\text{He}$ measurements throughout the Southern Ocean. 3-helium (^3He) is released to the deep ocean by hydrothermal vents along with dissolved iron. There it can be transported thousands of kilometers before being upwelled closer to the surface to support primary production (Riesing et al., 2015; Ardyna et al., 2019). Therefore, measurements of $\delta^3\text{He}$ can be used as a proxy for estimating the presence of hydrothermal iron. A transect of $\delta^3\text{He}$ along the 0° meridian from 60°S to 70°S (figure C1) gives a cross-section view of Maud Rise (approximately 65°S - 66°S). We noticed two $\delta^3\text{He}$ anomalies above 9 percent at the flanks of the seamount (68°S - 66°S and 65°S - 64°S) in the range of 200-900 m (figure C2). By looking at two additional transects of temperature and salinity, we find that the water masses, where the two $\delta^3\text{He}$ anomalies are, have a salinity in the range 34.50-34.75 PSS-78 and temperature above 1°C (figure C2). These water masses are warmer, saltier, higher in $\delta^3\text{He}$ and expectedly higher in dissolved iron concentration of hydrothermal origin than other water masses in the area

(20°W-20°E,50°S-70°S), hence supporting the hypothesis of iron supply from the deep ocean (measurements of these water masses are within the black rectangle in Figure C3).

Figure C1: **All stations with $\delta^3\text{He}$ measurements.** The boundaries of the region are (20°W-20°E,50°S-70°S). The red box encircled stations whom datasets are used for plotting the transects of Figure C2

Figure C2: $\delta^3\text{He}$ measurements and water masses identification. Transects along the 0° meridian from 60°S to 70°S of a) $\delta^3\text{He}$ b) temperature, c) salinity. Isolines are drawn in black.

Figure C3: **Temperature/Salinity diagram.** All $\delta^3\text{He}$ measurements of the GLODAP database in the region (20°W-20°E,50°S-70°S) are represented as dots with color varies according to the corresponding $\delta^3\text{He}$ value. Measurements of water masses with a salinity of ≈ 34.69 PSS-78 and a temperature of $\approx 1^\circ\text{C}$ are highlighted by a black rectangle.

Lastly, the paper that you reference when you first mention Weddell Deep Water (Lines 50-52), does not say anything about Weddell Deep Water, it is only about the warm water halo around Maud Rise. I think it would be wise to reference a paper that actually discusses this water mass and the properties associated with it. I found a few of these (in a search to satisfy my own curiosity) and also found that this water mass is usually referred to as Warm Deep Water, instead of Weddell Deep Water. I leave it up to you to decide on your naming conventions, but whatever paper you reference for a detailed description of this water mass should use the same name. I think including a reference that provides more details about this water mass is necessary as its properties are important to your conclusions, and many people will be looking for more information.

We thank the reviewer for pointing out this inconsistency. Weddell Deep Water has been replaced by Warm Deep Water throughout the text. We thank the reviewer for providing us with references about characteristics of the Warm Deep Water. We included these references in our text line 50.

Review Round 3

This paper describes the phytoplankton bloom that forms around Maud Rise in the eastern Weddell Sea. They link phytoplankton abundance to upwelling of warm/salty deep water layer due to the mesoscale dynamics on the edges of the Taylor column. They show the thinning of the winter water layer and the intrusion of the warm/salty deep water with glider, argo float, and ctd data. In addition to the previous addition showing the connection between changes in chlorophyll a concentration and changes in SST they have now added a figure showing the connection between changes in SST with changes in SSH.

The authors have addressed all of my previous comments to my satisfaction. I have one more general comment, and a few specific comments shown below. I believe these are all very easily handled with very minor changes to the manuscript.

We thank the reviewer for the positive and constructive comments about our paper. We respond below to all these new comments. We hope that both the editor and the reviewer will find this new version of the paper acceptable for publication in Nature Communication.

General comment:

In the section where you talk about the relevance of the Maud Rise Bloom and use the Argo float data to highlight how productive it is compared to other blooms in the Southern Ocean, you highlight that Maud Rise is responsible for 11 of the 39 profiles you find in the Argo record with integrated Chl over 100 mg m⁻³. Unless I am misunderstanding something this feels like an overstatement. Not that it isn't true, but what I understand from your paper is that these profiles are from the float that you deployed into these productive waters, while the other high chl profiles were encountered by chance. If I have misunderstood this, please make it clearer in your paper that this was not the float shown in Figure 1b that was specifically deployed into the bloom. If I have this correct, then it is inappropriate to play this up as you have introduced a sampling bias into the integrated chl data set by deploying a float into a known area of high chl and then comparing it to the frequency of other regions of high chl in the rest of the dataset. This needs to be made clear in your text. Either by clarifying that this data is from a different float or by acknowledging the sampling bias and addressing it directly in the text.

We thank the reviewer for pointing out this lack of information. The 39 profiles do not include the floats deployed over Maud Rise. We clarified this in the text lines 182-183 :

These 39 profiles do not include the 2 floats which were deployed over Maud Rise during the summer 2021-2022 to avoid any sampling bias.

Specific comments:

Figure 1 caption: You say that (d) and (e) show a longitudinal transect when it is clearly latitudinal, judging from the x-axis which shows it going from east to west. If you're looking to say this in a single word zonal or latitudinal work.

We apologize for this mistake. We actually meant zonal. This has been corrected.

Figure 4 caption: Add the definition that you use for MLD back to the caption. You had it there in a previous version, please put it back.

The MLD definition that we used was placed in the Methods section. We put it back at the end of the caption of Figure 4 as in the previous version.

Figure 4: You don't include the symbols above the sections that correspond to the symbols in the glider track maps. You had these in the last version. Please add them back.

We thank the reviewer for pointing out this problem. We changed the pixel resolution of the figure and during the process, the symbols went astray. This has been corrected as shown below :

Figure 4: **High resolution views of the halo by an underwater glider.** (a) Maximum temperature observed between 0-200 m during the glider survey. (b) Glider-observed integrated Chl-*a* concentration between 0-200 m. For both (a) and (b), isobaths are depicted every 200 m and the 3500 m isobath is shown in blue. Blue lines in the glider sections of (c) Chl-*a*, (d) temperature, and (e) salinity correspond to checkpoints marked by the symbols in a) and b). Isopycnals are depicted every 0.1 kg·m⁻³ between 1023.0 and 1030.0 kg·m⁻³. The thick black line in (c), (d), (e) is the mixed layer depth (MLD) based on the 0.03 kg m⁻³ density threshold criteria of De Boyer Montégut et al., 2004 [39].

Line 163: do you mean GLODAP instead of GLOBDATA?

We thank the reviewer for bringing this typo to our attention. We meant GLODAP. This has been corrected

Line 168 and Line 171: you left the “He” out of $\delta^3\text{He}$

We thank the reviewer for notifying us about this typo. It has been corrected.

Figure A4: Adjust the colorscale in Figure A4a to fit the range of the data which appear to go from -1 to 1 rather than -5 to 5.

Figure A4: Add hatching to indicate area where p-value was less than 0.05 on both A4 figures

We thank the reviewer for pointing out this inconsistency. We scaled back the colorscale of Figure A4a) to [-1:1] and we hatched the areas where the p-value is less than 0.05 on both Figures A4 a) and b).

Figure A4: **Slope of the correlation.** a) Slope of the Chl-*a* -SST correlation b) Slope of the SSH-SST correlation. Isobaths, plotted as black solid contours, are drawn every 500 meters. The 3500 meter isobath is drawn in Blue. Hatched areas show significant correlations ($p < 0.05$).

Review Round 2

This paper describes the phytoplankton bloom that forms around Maud Rise in the eastern Weddell Sea. They link phytoplankton abundance to upwelling of warm/salty deep water layer due to the mesoscale dynamics on the edges of the Taylor column. They show the thinning of the winter water layer and the intrusion of the warm/salty deep water with glider, argo float, and ctd data. They have also now linked changes in chlorophyll a concentration with changes in SST indicative of the upwelling of the warm deep water.

I very much want to see this paper published. I think the findings are very interesting. However, I still think there are issues that need to be addressed. I've detailed them in three sections below.

Stats reporting

The analysis of the relationship between chl and sst in Figure 3 does a good job of addressing my comments about interannual variability from the last round of reviews. However, there needs to be more information given on the relationship identified beyond just the correlation coefficient. I think it would be appropriate to report the p-value and the slope of the relationship. I don't expect that the p-value would tell a very different story than the correlation coefficient already presented, but it should be included. The slope gives us additional information on the magnitude of the relationship not just the strength of it. How much does the chl change with increasing or decreasing SST? I would recommend adding these to the supplemental and referencing them briefly in the text.

I also feel like you are leaving something on the table by not taking your analysis further by now looking at the relationship between SSH and SST, or chl and SSH. It would be a very powerful connection if there is in fact a significant correlation between either of these variables and SSH. Then you tie the strength of the upwelling (and the bloom) to the strength of the Taylor column. I'm not going to insist on it, but it would be an excellent addition to your narrative. And seemingly with all of the data you have put together and the code you have presumably already developed, it should be a relatively easy analysis.

Data availability

You need to make the data that you used in your analyses available publicly. In my last review I asked for the publication of the Argo data that you used to make the maps in Figure A4, and you responded with a paragraph of text about satellite data. Clearly this was a mistake, however the data still have not been published.

I mocked up a version of the same figure as A4 from some float profiles that I have for the Pacific sector of the SO that shows all of the profiles with integrated chl (down to 200 meters) above 100 mg m⁻². There are a lot more than what you show in your figure A4. Out of 1227 profiles I have 146 with integrated chl above 100 mg m⁻² (11%) versus the 39 out of 8560 that you found (0.4%).

It's not that I don't believe your results, it's that I don't think you provide enough information for me to get the same answer. You hang the importance of the Maud Rise bloom on the magnitude relative to other blooms in the Southern Ocean, at a minimum you need to publish these data, and you should also add more detail on the chl fluorescence corrections used. If the version of the Argo chl data that you used is a publicly available product, then link to that. But the link to the Argo website and the original (presumably uncorrected) float data results in the answer that I show in the figure below. I need to understand why it's different and how to get the answer that you get, and then you need to provide the data.

I didn't ask for you to publish the satellite data in my last review, because you were only using them for descriptive purposes, but now you are running statistical analyses and you need to put the data that you used in a public repository. This includes the data for the images in figures A1 and A2 (the December averages for chl and sst for each year) as well as the data used to calculate the sea ice and SSH climatologies in Figure 2. Linking to the website that provides the daily satellite images is not enough. This would require someone to download daily images for the relevant time period for each year (~775 images for each of 4 different satellite products) and then create averages for each year, just to evaluate your findings. You do state that code to make the figures is available upon request. This is not the current standard for publication. The data need to be publicly available. You should probably include the glider data too. Referencing the float numbers and website for the float data that you used seems fine as those are curated in a way that makes them very easy to access.

Deep iron concentrations

In reading through the comments from Reviewer 1, my attention was drawn to the source of the iron being delivered from the deep water, and I went down a bit of a rabbit hole and found several issues. As reviewer 1 identified, you do not really resolve the issue of whether the water mass being upwelled does in fact have enough Fe to generate the observed bloom. And the changes that you made in response to their comment, do not really address this issue.

This needs to be addressed but I think it is a simple fix. In your response to reviewer 1 you mention hydrothermal Fe and the elevated ^3He found in the eastern Weddell, but you never mention findings on this in the paper and only reference papers that say this without ever mentioning the word hydrothermal. I have read the Moreau paper that you reference previously, but in looking back at it, one of the transects that measures ^3He , practically runs over Maud Rise. You can directly address the origin of the Fe by plotting a section of the ^3He from this transect and then do a water mass analysis that shows that the elevated ^3He is in fact associated with the WDW water mass. This was the method used to link phytoplankton blooms near the SWIR to hydrothermal vents upstream in the Ardyna paper that you also reference, and would satisfy me that we could reasonably expect elevated Fe concentrations in the WDW water mass.

To satisfy my own curiosity and to not be that reviewer that sends you down a horribly unproductive blind alley, I took a look at the ^3He data for the transect in question (shown below; bottom left shows the map with the ^3He transect shown in the top right outlined in red, top left shows a map of the maximum ^3He value for each profile in color, and bottom right is a TS plot with ^3He shown in color). You can actually see in the section plot where the transect passes over the shoulder of Maud Rise at 65°S and there is clearly an elevated ^3He anomaly there. This should be a pretty straightforward connection to make simply by referencing the T/S range of your warm/salty water mass and mapping it onto the T/S space for the transect.

Lastly, the paper that you reference when you first mention Weddell Deep Water (Lines 50-52), does not say anything about Weddell Deep Water, it is only about the warm water halo around Maud Rise. I think it would be wise to reference a paper that actually discusses this water mass and the properties associated with it. I found a few of these (in a search to satisfy my own curiosity) and also found that this water mass is usually referred to as Warm Deep Water, instead of Weddell Deep Water. I leave it up to you to decide on your naming conventions, but whatever paper you reference for a detailed description of this water mass should use the same name. I think including a reference that provides more details about this water mass is necessary as its properties are important to your conclusions, and many people will be looking for more information.

Hutchinson, K., Deshayes, J., Sallee, J. B., Dowdeswell, J. A., de Lavergne, C., Ansorge, I., ... & Fawcett, S. E. (2020). Water mass characteristics and distribution adjacent to Larsen C Ice Shelf, Antarctica. *Journal of Geophysical Research: Oceans*, 125(4), e2019JC015855.

Robertson, R., Visbeck, M., Gordon, A. L., & Fahrbach, E. (2002). Long-term temperature trends in the deep waters of the Weddell Sea. *Deep Sea Research Part II: Topical Studies in Oceanography*, 49(21), 4791-4806.